# Association of Coffee and Tea Consumption with the Risk of Asthma: A Prospective Cohort Study from the UK Biobank

**DOI:** 10.3390/nu14194039

**Published:** 2022-09-28

**Authors:** Fengyu Lin, Yiqun Zhu, Huaying Liang, Dianwu Li, Danrong Jing, Hong Liu, Pinhua Pan, Yan Zhang

**Affiliations:** 1Center of Respiratory Medicine, Xiangya Hospital, Central South University, Changsha 410008, China; 2National Key Clinical Specialty, Branch of National Clinical Research Center for Respiratory Disease, Xiangya Hospital, Central South University, Changsha 410008, China; 3Hunan Engineering Research Center for Intelligent Diagnosis and Treatment of Respiratory Disease, Changsha 410008, China; 4Department of Dermatology, Xiangya Hospital, Central South University, Changsha 410008, China; 5National Clinical Research Center for Geriatric Disorders, Xiangya Hospital, Central South University, Changsha 410008, China

**Keywords:** asthma, coffee, tea, caffeine, UK Biobank

## Abstract

Background: Previous observational studies investigated the relationship between coffee and tea intake and the risk of asthma, however, the conclusions were inconsistent. Further, the combined effect of coffee and tea consumption on asthma has rarely been studied. Methods: We examined associations between the self-reported intake of tea and coffee and the risk of incident asthma in a total of 424,725 participants aged from 39 to 73 years old from the UK Biobank. Cox proportional hazards models were used to estimate the associations between coffee/tea consumption and incident adult-onset asthma, adjusting for age, sex, race, smoking status, body mass index (BMI), education, and Townsend deprivation index. Results: Cox models with penalized splines showed J-shaped associations of coffee, tea, caffeinated coffee, and caffeine intake from coffee and tea with the risk of adult-onset asthma (*p* for nonlinear <0.01). Coffee intake of 2 to 3 cups/d (hazard ratio [HR] 0.877, 95% confidence interval [CI] 0.826–0.931) or tea intake of 0.5 to 1 cups/d (HR 0.889, 95% CI 0.816–0.968) or caffeinated coffee intake of 2 to 3 cups/d (HR 0.858, 95% CI 0.806–0.915) or combination caffeine intake from tea and coffee of 160.0 to 235.0 mg per day (HR 0.899, 95% CI 0.842–0.961) were linked with the lowest hazard ratio of incident asthma after adjustment for age, sex, race, smoking status, BMI, qualification, and Townsend deprivation index. Conclusions: Collectively, the study showed light-to-moderate coffee and tea consumption was associated with a reduced risk of adult-onset asthma and controlling total caffeine intake from coffee and tea for a moderate caffeine dose of 160.0 to 305.0 mg/day may be protective against adult-onset asthma. Further investigation on the possible preventive role of caffeine in asthma is warranted.

## 1. Introduction

Asthma is a common chronic respiratory disease that is characterized by airway hyper-responsiveness, inflammation, and remodeling, and it affects more than 339 million people worldwide, which causes a large health and economic burden [1]. Although multiple risk factors for asthma have been identified over the past 30 years, little progress has been made in exploring protective factors for preventing asthma [2].

Coffee and tea are the most popularly consumed beverages worldwide, thus, any biological effects from coffee and tea consumption could have a significant influence on public health [3]. It has been reported that coffee reduces the risks of multiple cancers [4], cardiovascular disease [5], diabetes [6], stroke [7], Parkinson’s disease [8], and gallstones [9]. Similarly, tea has also been proposed to exert a protective role in various cardiopulmonary conditions or other chronic diseases via multiple biologic functions, such as antioxidation, anti-inflammation, and immuno-regulation [10].

Several studies investigated the relationship between coffee and tea consumption and the prevalence and incidence of asthma; however, the conclusions were not consistent. Cross-sectional studies suggested that coffee consumption may link to a low prevalence of asthma [11,12]. Chronic coffee consumption appeared to be associated with improved asthma control [13]. Moreover, another study found an association between tea consumption and prevalence/incidence of asthma in the crude model; however, this association became statistically insignificant in the adjusted model [12]. In addition, a recent cross-sectional/prospective cohort study also found no significant association between coffee or tea consumption and asthma prevalence [14]. Collectively, the above studies with conflicting results lead to the uncertainty of the relationship, and whether asthma benefits from coffee and tea intake remains inconclusive [15].

Coffee and tea are different types of beverages, which contain functional components with both common and distinct contents [16,17,18]. The caffeine is highly enriched in coffee and moderately enriched in tea. Previous studies have reported that the positive effects of coffee are mainly related to caffeine, as its metabolites have greater exposure to tissues than others [19]. Remarkably, caffeine is known as a central nervous system stimulant and bronchodilator and can slightly improve lung function in adults. Furthermore, caffeine could metabolize to theophylline, which is still one of the most widely prescribed drugs for the treatment of asthma [20,21], thus proposing a protective role of caffeine in asthma. However, caffeine at very high levels of intake may also cause multiple side effects that affect asthma in a negative way, such as insomnia, nervousness, reduced quality of sleep, and increased anxiety. Therefore, the possible benefits of caffeine content in coffee and tea drinking must be weighed against potential risks, and a large prospective cohort study to assess the relationship between caffeine and asthma is warranted.

Thus, we used the large population-based cohort study from the UK Biobank to examine the individual effect of coffee- and tea-drinking behaviors on the risk of developing adult-onset asthma, and the effect of caffeine intake in combining coffee and tea consumption on asthma incidence as well.

## 2. Methods

### 2.1. Study Design and Population

We used the large-scale cohort study from the UK Biobank that recruited more than 500,000 participants aged 37–73 years from 2006–2010 [22,23]. Prospective participants were identified from National Health Service registers in the United Kingdom, and then were invited to one of 22 assessment centers to fill out a baseline questionnaire about epidemiologic data, undergo a physical examination, and donate biological samples. The UK Biobank was conducted in accordance with the Declaration of Helsinki, and the protocol has already received ethical approval from the Northwest Multi-Center Research Ethics Committee (11/NW/0382) and obtained written informed consent from all participants. Participants with baseline asthma and participants with missing data on coffee or tea intake, sex, age, smoking, or BMI were excluded from the analysis. Data from 424,725 individuals were available for analysis in our present study.

### 2.2. Exposure Assessment

Coffee and tea intake were assessed by applying questionnaires during the recruitment. Participants were asked the following questions: “How many cups of coffee do you drink each day (including decaffeinated coffee)?”; “How many cups of tea do you drink each day (including black and green tea)?” Participants would select the choice of the number of cups, “Less than one,” “Do not know,” or “Prefer not to answer.” Further, coffee drinkers were further asked the type of coffee they usually drank with the answers of “Decaffeinated coffee (any type)”, “Instant coffee”, “Ground coffee (including espresso and filtered coffee), “other type of coffee”, “Do not know”, or “prefer not to answer”. Caffeine intake (mg/day) was calculated based on self-reported coffee and tea intake from the questionnaire and assumed one cup of coffee to 75 mg of caffeine and one cup of tea as 40 mg of caffeine [24,25,26].

### 2.3. Ascertainment of Asthma

Asthma events were recorded by the International Classification Disease (ICD-10) coding system: asthma (ICD codes: J45). Incident death events were identified by death registry records. Follow-up began at recruitment and lasted until asthma was diagnosed, death, loss of follow-up, or December 2020.

### 2.4. Covariates

In the present study, we considered the following potential covariates: age, gender, race, smoking status (never, previous, and current), body mass index (BMI), an education level (college or university degree, professional qualifications, A Levels/AS Levels or equivalent, O Levels/GCSEs or equivalent, and none of the above), Townsend deprivation index. All measures were obtained using the baseline questionnaire.

### 2.5. Statistical Analyses

The baseline characteristics were summarized across coffee, tea, and combined caffeine intake. Summary statistics for categorical variables are described as frequencies (percentages) and analyzed using a chi-squared test to study differences in population. Continuous variables were expressed as means (SD) and were compared by analysis of the student’s t-test. Restricted cubic spline models were used to assess potential nonlinear associations between coffee, tea, or combined caffeine intake per day with the risk of incident asthma. Those who drank coffee or tea beyond 15 cups per day were recorded as 15 cups/day. Multiple imputations were performed to recover missing data. Cox proportional hazards regression models were also used to investigate the association between coffee, tea, and combined caffeine intake with the risk of incident asthma. Coffee and tea intake were classified into 0, 0.5 to 1, 2 to 3, and ≥4 cups per day. Total caffeine intake of tea and coffee was grouped by quartile into Q1 (≤160.0 mg/day), Q2 (160.0–235.0 mg/day), Q3 (235.0–305.0 mg/day), and Q4 (305.0–390.0 mg/day). All models were adjusted for age, sex, BMI, education level, and Townsend deprivation index, and adjusted for tea in coffee analysis or for coffee in tea analysis. Several sensitivity analyses were performed to check the robustness of the results. Subgroup analyses were stratified by age (<60 versus ≥60 years), sex (male versus female), BMI (<25, 25 to <30 versus ≥30 kg/m^2^), and smoking status (never, former versus current). Furthermore, we calculated the interaction between coffee, tea, and or combined caffeine intake and subgroups to evaluate the heterogeneity of this association across different levels of these covariates. *p* values less than 0.05 were statistically significant. Analyses were performed using SPSS version 26.0 (IBM Corporation, Armonk, NY, USA), GraphPad Prism version 9.0 software (GraphPad Software Inc., San Diego, CA, USA), or R-4.1.2 software (R Foundation for Statistical Computing, Vienna, Austria).

## 3. Results

### 3.1. Baseline Characteristics

Of the 504,364 baseline participants, 424,725 participants were ultimately included in the analyses (Figure 1). Of 424,725 participants, the mean age was 56.61 ± 8.06 years, and 195,568 (46.0%) were male. Overall, 93,503 (22.0%) were non-coffee drinkers and 61,993 (14.6%) were non-tea drinkers. In total, 331,222 (78.0%) participants were coffee drinkers, with 27.3% of participants reporting reported drinking 0.5 to 1 cup of coffee, 31.2% drinking 2 to 3 cups, and 19.4% drinking ≥4 cups per day. A total of 362,732 (85.4%) participants were tea drinkers, with 11.5% of participants reporting reported drinking 0.5 to 1 cup of tea, 29.5% drinking 2 to 3 cups, and 44.4% drinking ≥4 cups per day. During the follow-up period, 8680 participants (2.0%) developed asthma, and the median time to development of the disease was 5.0 years (range from 3.0 to 7.0 years). Non-coffee drinkers and participants drinking 0.5 to 1 cup of coffee reported higher cups of tea consumption, while non-tea drinkers and participants drinking 0.5 to 1 cup of tea reported higher cups of coffee consumption. Compared to participants drinking 0.5 to 1 or 2 to 3 cups of tea, participants drinking ≥4 cups of tea were more likely to drink decaffeinated coffee (Table 1). To statistically assess the association between daily caffeine intake from coffee and tea with asthma incidence, we then grouped participants in five quartiles by tea and coffee combined caffeine intake. Participants in quartile 2 (take caffeine 160.0–235.0 mg/day) have the lowest incidence rate ratios for asthma (Table 1).

### 3.2. Nonlinear Association

Restricted cubic spline models were used to evaluate the relationship between coffee, tea, and combined caffeine intake with asthma. Associations between coffee, tea, and combined caffeine intake with risk of asthma were nonlinear (*p* for nonlinear <0.001), both in unadjusted (Figure 2A) and multiple-adjusted models (Figure 2B). Significant decreases in risk were observed for asthma at light-to-moderate coffee and tea intake, but not at none and high intake. In addition, the low intake of tea and coffee combined with caffeine intake was associated with a lower risk of asthma, while the high intake was not. Additionally, a similar J-shaped association was observed for coffee intake, tea intake, caffeinated coffee intake, and combined caffeine intake from coffee and tea (Figure 2).

### 3.3. Coffee Intake, Tea Intake, and Combined Caffeine Intake with Asthma Risk

Univariate and multivariable cox regression analyses showed that compared with noncoffee participants, those who drank 0.5 to 1 (adjusted HR = 0.919, 95%CI 0.866~0.976, *p* = 0.006) or 2 to 3 cups (adjusted HR = 0.877, 95%CI 0.826~0.931, *p* < 0.001) coffee per day had a lower risk of asthma. Compared with participants who drank non-tea, participants with 0.5 to 1 (adjusted HR = 0.889, 95%CI 0.816~0.968, *p* < 0.001) or 2 to 3 cups (adjusted HR = 0.929, 95%CI 0.868~0.995, *p* < 0.037) of tea per day also had a lower risk of asthma. Further cox regression analyses were taken in caffeinated coffee consumption to assess the effect of caffeine, the results showed that the HR for asthma was reduced slightly both in participants with 0.5 to 1 (adjusted HR = 0.883, 95%CI 0.829~0.941, *p* < 0.001) and 2 to 3 cups (adjusted HR = 0.858, 95%CI 0.806~0.915, *p* < 0.001), compared with total coffee. (Table 2; Figure 3). However, participants drinking ≥4 cups of coffee, tea, or caffeinated coffee per day had no decreased risk of asthma. Furthermore, 160.0–235.0 mg/day of combination caffeine intake from coffee and tea intake was associated with a lower risk of asthma (adjusted HR = 0.899, 95%CI 0.842~0.961, *p* = 0.002) (Table 2; Figure 3).

### 3.4. Subgroup and Sensitivity Analyses

Participants with 160.0–235.0 mg/day of combination caffeine intake with the risk of asthma were more pronounced in individuals aged lower than 60 years old and females. However, participants under 60 years and the male subgroup showed an inverse trend without significance between tea intake and combination caffeine intake with asthma risk for low-to-moderate consumption of tea and combination caffeine intake (Figure 4).

Subgroup analyses were conducted according to age, sex, and smoking status (All *p* values for interaction < 0.001) (Appendix A). In subgroup analyses for all types of coffee and caffeinated coffee consumption, except for the current smoking group, all the others showed a lower risk of incident asthma at a daily consumption level of 0.5 to 1 and/or 2 to 3 cups of coffee (Figure 4A,C). Participants with 160.0–235.0 mg/day of combination caffeine intake with the risk of asthma were more pronounced in individuals aged lower than 60 years old (adjusted HR = 0.866, 95%CI 0.789~0.951, *p* = 0.003), females (adjusted HR = 0.855, 95%CI 0.785~0.930, *p* < 0.001), and non-smokers (adjusted HR = 0.867, 95%CI 0.792~0.949, *p* = 0.002) (Figure 4D). However, participants under 60 years and the male subgroup showed no association between tea intake and combination caffeine intake with asthma risk, although an HR of less than 1 was observed for low-to-moderate consumption of tea and combination caffeine intake (Figure 4B,D). Appendix A presented results from our sensitivity analyses. Similar results were observed when we repeated our analyses in a subset with complete covariates data (Appendix A). Moreover, the results were barely changed after excluding participants with incident asthma during the first 2 years of follow-up (Appendix A).

## 4. Discussion

In this prospective cohort study based on the UK Biobank, we found a J-shaped association was observed between coffee, tea intake, caffeinated coffee intake, and caffeine intake with the risk of adult-onset asthma. Specifically, we demonstrated that light-to-moderate coffee and tea separately (0.5 to 3 cups per day) and 160.0 to 305.0 mg/day of combination caffeine intake from coffee and tea were associated with reductions in the risk of incident asthma, compared with nonhabitual drinkers.

Despite the growing concerns arising about the association of coffee and tea intake with health benefits in recent years, limited attention has been played to coffee and tea intake and the risk of asthma. Most studies investigating the relationship between coffee and tea consumption with asthma incidence were stuck 20 years ago with inconsistent results [11,13,14]. Although a new study of 173,209 participants and 3146 cases from Korea reported a relationship between coffee and tea intake and the prevalence of asthma, the cross-sectional design was not sufficient to infer causality [12]. To date, large longitudinal cohort studies were lacking to determine the relationship between coffee and tea intake and asthma. Herein, we observed a J relationship between coffee and tea intake and adult-onset asthma in a large prospective study, which has not been previously reported. Specifically, only regularly light-to-moderate coffee drinkers (0.5 to 3 cups per day) could significantly reduce the risk of adult-onset asthma incidence. The protective effects disappeared in heavy drinkers of coffee and tea in the study, which was inconsistent with the study of Wee et al. [12]. However, the beneficial J-shaped effect of coffee and tea intakes has been extensively reported in other diseases with large prospective cohort studies, such as stroke, dementia [7], and cancer mortality [27].

In addition, we also found that total caffeine intake from coffee and tea may play a central role in the association between coffee and tea consumption and asthma incidence. Caffeine is known to be a common ingredient in bronchodilators that could improve airway function modestly in asthmatics [19,28]. The protective role of caffeine intake may rely on its ability to relax respiratory muscles [29] and its anti-inflammatory and antioxidant properties [28]. Additionally, caffeine and catechin could synergistically potentiate the mast cell-stabilizing property as well [17], and more studies about specific mechanisms of caffeine on asthma are warranted. In the study, a similar J-shaped effect of caffeine intake against asthma was observed as well. Previous studies revealed that a single dose caffeine intake of 200 mg or less is safe, while a caffeine intake over 300 mg could result in toxic effects, such as restlessness, increased urination, gastrointestinal disorders, arrhythmia, and psychomotor agitation [30]. In most cases, a dose caffeine intake of less than 400 mg/day in healthy adults has been suggested [31,32]. Meanwhile, our study also shows the protective dose of caffeine intake from coffee and tea with asthma incidence in the range of 160.0 to 305.0 mg/day and suggests avoiding heavy caffeine consumption.

In the subgroup analysis, age, gender, BMI, and smoking status were noted to have a certain impact on these inverse associations with caffeine intake and asthma incidence. On the one hand, the possible reasons for those differences may be the complex effect on caffeine absorption and caffeine metabolism in heterogeneous participants, the greatest effect may be observed in CYP1A2 activity, a major enzyme that metabolizes caffeine (in about 80%) [33]. Herein, the effect of caffeine consumption from coffee and tea was significant in the females but not in the males. The sex discrepancy was also reported in a previous study showing that coffee consumption was inversely associated with asthma in women but not in men [12], which might be explained by the faster caffeine clearance in males than females due to the inhibited CYP1A2 activity by estradiol. In addition, smoking was also reported to be associated with faster caffeine metabolism and heavy coffee drinking because of the higher CYP1A2 activity in smokers [34]. Our results were significantly affected by the smoking status that the ideal dosage of habitual caffeine consumption from coffee and tea against asthma in former smokers was high doses (305.0 to 390.0 mg/kg per day) and presented no protective effect or even detrimental effect on the incidence of asthma in current smokers. On the other hand, asthmatics could also be influenced by a wide variety of factors, including age [35,36], sex [37], BMI [38,39], and smoking status [40]. In addition to the factors previously discussed, educational background and Townsend deprivation index are socioeconomic factors that exert effects on environmental allergen exposure [41,42] and may lead to variations in coffee and tea consumption [43,44]. Similar to other studies [7,25,45], we adjusted for education and Townsend deprivation index in this study. Although we found coffee and tea intakes were associated with a lower incidence of asthma in the large prospective cohort study, confirming the preventive and therapeutic role of coffee and tea intakes still warrants large well-designed clinical trials.

### Strengths and Limitations

This study is the first large prospective cohort study with sufficient follow-up to investigate the relationship between coffee and tea intake with adult-onset asthma; however, there are also certain limitations. Firstly, most participants in UK Biobank were of European ancestry, which would restrict the interpretation of the findings to other ethnic background populations. Secondly, coffee and tea intakes were self-reported based on questionnaires, and the changes in coffee and tea consumption over time were not evaluated after the baseline evaluation. Thirdly, caffeine intake levels were assessed on the self-report coffee and tea intake using the conversion equation provided in the literature without considering some potential confounders, such as chocolate, energy drinks, soft drinks, and medications intake [46,47]. Therefore, further validated tests, containing a precisely controlled content of caffeine, are needed to verify our conclusions. Additionally, the observed associations were slightly modified by genetic factors related to caffeine metabolism in previous studies [25]; however, we did not evaluate the effect of genetic variation in the study. Finally, the ages of participants ranged from 37 to 73 years old, and thus the findings should be deliberately interpreted into children- and adolescent-onset asthma.

## 5. Conclusions

In conclusion, we found that light-to-moderate coffee and tea consumption (0.5 to 3 cups per day) was associated with a reduced risk of adult-onset asthma and controlling total caffeine intake from coffee and tea for a moderate caffeine dose of 160.0 to 305.0 mg/day may be protective against adult-onset asthma.

## Figures and Tables

**Figure 1 nutrients-14-04039-f001:**
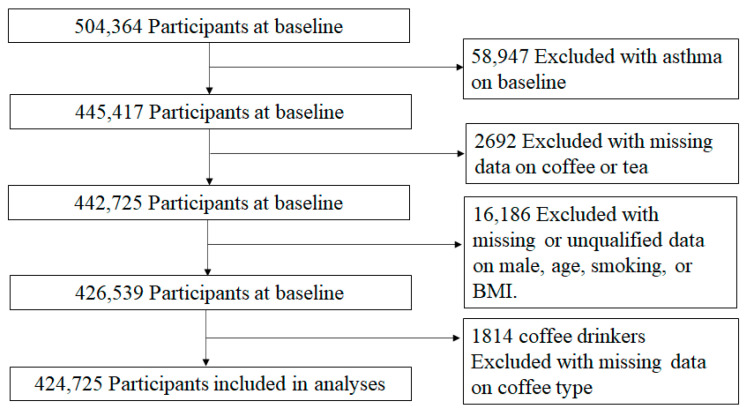
Flowchart of participant selection.

**Figure 2 nutrients-14-04039-f002:**
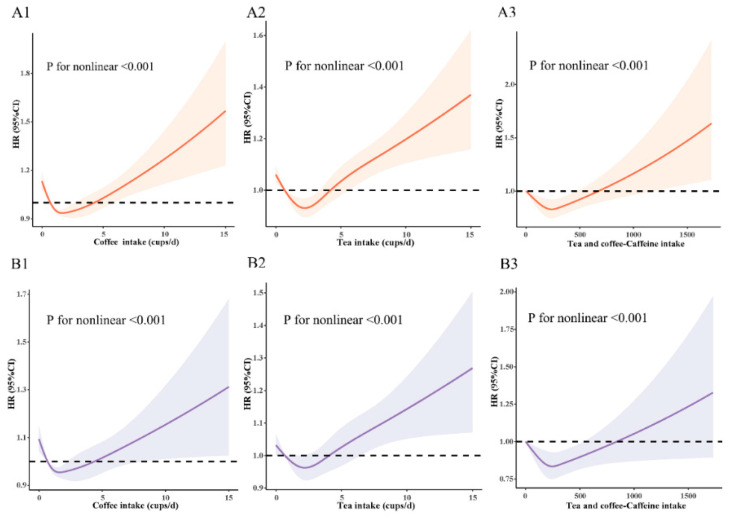
Restricted cubic spline models for the relationship between coffee, tea, and combination caffeine intake from coffee and tea with adult-onset asthma. (**A**) The restricted cubic spline model is unadjusted: (**A1**) Coffee and adult-onset asthma; (**A2**) Tea and adult-onset asthma; (**A3**) combination caffeine intake from coffee and tea and adult-onset asthma. (**B**) Restricted cubic spline model is adjusted by age, gender, race, BMI, smoking status, education, and Townsend Index and adjusted for coffee in tea analysis or for tea in coffee analysis: (**B1**) Coffee and adult-onset asthma, (**B2**) Tea and adult-onset asthma, (**B3**) combination caffeine intake from coffee and tea and adult-onset asthma. The 95% CIs of the adjusted HRs are represented by the shaded area. BMI, body mass index; HR: hazard ratio; CI, confidence interval.

**Figure 3 nutrients-14-04039-f003:**
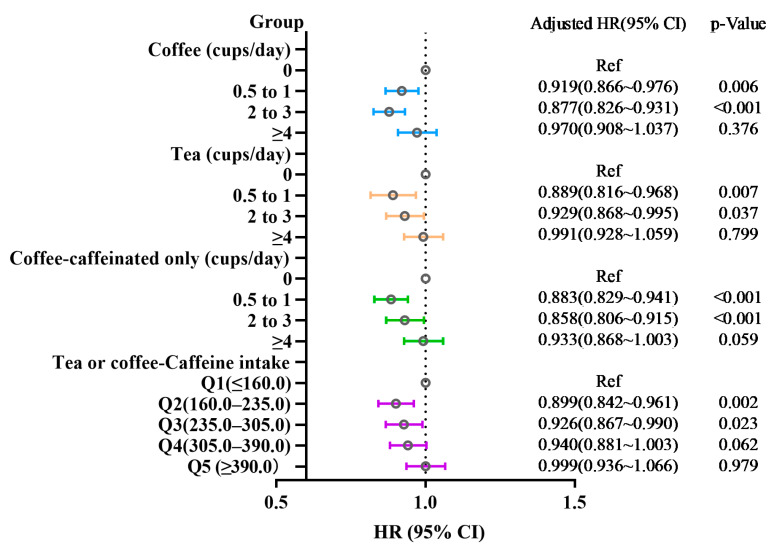
Summary of hazard ratios and 95% confidence intervals (*x*-axis) for cups per day categories and linear trend for coffee, tea, caffeinated coffee, and combination caffeine intake from coffee and tea for adult-onset asthma. The model is adjusted by age, gender, race, BMI, smoking status, education, and Townsend Index, and adjusted for coffee in tea analysis or for tea in coffee analysis. Hatched vertical line indicates the null reference value of ‘1′.

**Figure 4 nutrients-14-04039-f004:**
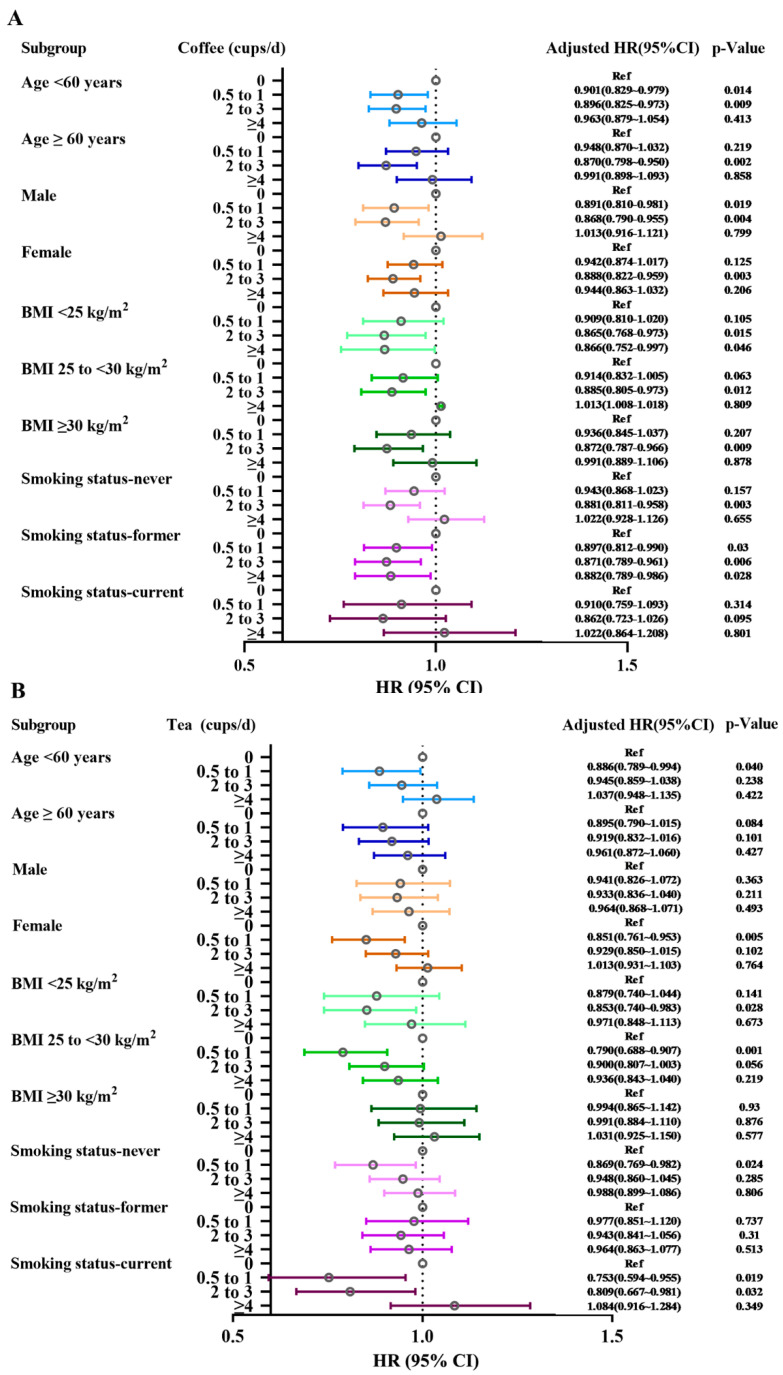
Subgroup analyses according to age, gender, BMI, and smoking status. Summary of hazard ratios and 95% confidence intervals (*x*-axis) for cups per day categories and linear trend for (**A**) coffee, (**B**) tea, (**C**) caffeinated coffee, and (**D**) combination caffeine intake from coffee and tea for adult-onset asthma. The model is adjusted by age, gender, race, BMI, smoking status, education, and Townsend Index and adjusted for coffee in tea analysis or for tea in coffee analysis. Hatched vertical line indicates the null reference value of ‘1′.

**Table 1 nutrients-14-04039-t001:** Baseline characteristics by coffee and tea intake in the UK Biobank cohort.

		Coffee Intake, cups/day, No. (%)				Tea Intake, cups/day, No. (%)				Caffeine Intake, mg/day, No. (%)				
Characteristic	All Participants	0	0.5 to 1	2 to 3	≥4	0	0.5 to 1	2 to 3	≥4	Q1(≤160.0)	Q2 (160.0–235.0)	Q3 (235.0–305.0)	Q4 (305.0–390.0)	Q5 (≥390.0)
Total number, No. (%)	424,725	93,503 (22.0)	115,909 (27.3)	132,751 (31.3)	82,562 (19.4)	61,993 (14.6)	48,852 (11.5)	125,407 (29.5)	188,473 (44.4)	93,308 (22.0)	84,391 (19.9)	81,117 (19.1)	86,317 (20.3)	79,592 (18.7)
Asthma cases, No. (%)	8680 (2.0)	2130 (2.3)	2305 (2.0)	2476 (1.9)	1769 (2.1)	1371 (2.2)	889 (1.8)	2417 (1.9)	4003 (2.1)	1991 (2.1)	1602 (1.9)	1597 (2.0)	1764 (2.0)	1757 (2.2)
Age, mean (SD), y	56.61 (8.06)	55.52 (8.21)	57.09 (8.02)	57.16 (7.97)	56.30 (7.95)	55.52 (8.22)	55.60 (8.32)	56.75 (8.10)	57.14 (7.85)	55.29 (8.39)	57.00 (8.10)	57.11 (7.90)	57.34 (7.79)	56.45 (7.88)
Gender														
Male, n (%)	195,568 (46.0)	39,726 (42.5)	50,681 (43.7)	62,304 (46.9)	42,857 (51.9)	27,370 (44.2)	23,580 (48.3)	57,911 (46.2)	86,707 (46.0)	39,342 (42.2)	37,471 (44.4)	36,365 (44.8)	40,319 (46.7)	42,071 (52.9)
Female, n (%)	229,157 (54.0)	53,777 (57.5)	65,228 (56.3)	70,712 (53.1)	39,705 (48.1)	34,623 (55.8)	25,272 (51.7)	67,496 (53.8)	101,766 (54.0)	53,966 (57.8)	46,920 (55.6)	44,725 (55.2)	45,998 (53.3)	37,521 (47.1)
Race														
White, n (%)	385,709 (90.8)	82,724 (88.5)	104,622 (90.3)	121,406 (91.5)	76,957 (93.2)	56,422 (91.0)	42,401 (86.8)	112,582 (89.8)	174,304 (92.5)	79,931 (85.7)	76,349 (90.5)	74,983 (92.4)	80,425 (93.2)	74,021 (93.0)
Non-white, n (%)	39,016 (9.2)	10,779 (11.5)	11,287 (9.7)	11,345 (8.5)	5605 (6.8)	5571 (9.0)	6451 (13.2)	12,825 (10.2)	14,169 (7.5)	13,377 (14.3)	8042 (9.5)	6134 (7.6)	5892 (6.8)	5571 (7.0)
Coffee intake, mean (SD)	2.02 (2.07)	0 (0)	0.87 (0.22)	2.39 (0.49)	5.30 (2.03)	3.47 (2.77)	2.78 (2.11)	1.97 (1.67)	1.37 (1.69)	0.50 (0.63)	1.33 (0.91)	1.70 (1.37)	2.42 (1.39)	4.42 (2.88)
Tea intake, mean (SD)	3.42 (2.86)	4.52 (3.29)	4.03 (2.59)	2.97 (2.36)	2.03 (2.72)	0 (0)	0.87 (0.22)	2.51 (0.50)	5.81 (2.54)	1.80 (1.40)	2.81 (1.60)	3.60 (2.23)	4.11 (2.33)	5.02 (4.59)
Smoking status														
Never, n (%)	232,699 (54.8)	55,079 (58.9)	67,175 (58.0)	72,692 (54.8)	37,753 (45.7)	32,324 (52.1)	26,598 (54.4)	71,033 (56.6)	102,744 (54.5)	57,178 (61.3)	47,934 (56.8)	45,268 (55.8)	46,339 (53.7)	35,980 (45.2)
Previous, n (%)	147,129 (34.6)	29,361 (31.4)	40,004 (34.5)	47,618 (35.9)	30,146 (36.5)	21,078 (34.0)	16,857 (34.5)	43,786 (34.9)	65,408 (34.7)	29,346 (31.5)	30,054 (35.3)	28,649 (35.3)	30,854 (35.7)	28,226 (35.5)
Current, n (%)	44,897 (10.6)	9063 (9.7)	8730 (7.5)	12,441 (9.4)	14,663 (17.8)	8591 (13.9)	5397 (11.0)	10,588 (8.4)	20,321 (10.8)	6784 (7.3)	6403 (7.6)	7200 (8.9)	9124 (10.6)	15,386 (19.3)
Body mass index, mean (SD)	27.31 (4.69)	27.37 (4.90)	26.96 (4.60)	27.19 (4.53)	27.92 (4.76)	27.97 (5.16)	27.32 (4.78)	27.10 (4.57)	27.23 (4.56)	27.26 (4.99)	27.07 (4.61)	27.20 (4.57)	27.31 (4.50)	27.74 (4.70)
BMI category, n (%)														
<25	142,884 (33.6)	31,945 (34.2)	42,783 (36.9)	45,142 (34.0)	23,014 (27.9)	18,738 (30.2)	16,653 (34.1)	44,083 (35.2)	63,410 (33.6)	33,631 (36.0)	30,291 (35.9)	27,604 (34.0)	28,125 (32.6)	23,233 (29.2)
25 to <30	181,689 (42.8)	38,480 (41.1)	48,638 (42.0)	57,957 (43.7)	36,614 (44.3)	25,301 (40.8)	20,541 (42.0)	53,782 (42.9)	82,065 (43.5)	37,265 (39.9)	35,643 (42.2)	35,294 (43.5)	38,153 (44.2)	35,334 (44.4)
≥30	100,152 (23.6)	23,078 (24.7)	24,488 (21.1)	29,652 (22.3)	22,934 (27.8)	17,954 (29.0)	11,658 (23.9)	27,542 (22.0)	42,998 (22.8)	22,412 (24.0)	18,457 (21.9)	18,219 (22.5)	20,039 (23.2)	21,025 (26.4)
Education														
College or University Degree	137,939 (32.5)	25,112 (26.9)	39,162 (33.8)	47,804 (36.0)	25,861 (31.3)	18,838 (30.4)	19,577 (40.1)	43,705 (34.9)	55,819 (29.6)	30,410 (32.6)	28,631 (33.9)	26,548 (32.7)	28,291 (32.8)	24,059 (30.2)
Professional Qualifications	50,080 (11.8)	11,371 (12.2)	13,369 (11.5)	15,054 (11.3)	10,286 (12.5)	7296 (11.8)	4937 (10.1)	14,061 (11.2)	23,786 (12.6)	10,226 (11.0)	9514 (11.3)	9531 (11.7)	10,631 (12.3)	10,178 (12.8)
A Levels/AS Levels or Equivalent	475,04 (11.2)	9799 (10.5)	13,452 (11.6)	15,247 (11.5)	9006 (10.9)	7185 (11.6)	6121 (12.5)	14,241 (11.4)	19,957 (10.6)	10,820 (11.6)	9583 (11.4)	9207 (11.4)	9460 (11.0)	8434 (10.6)
O Levels/GCSEs or Equivalent	113,778 (26.8)	26,311 (28.1)	30,201 (26.1)	34,284 (25.8)	22,982 (27.8)	17,969 (29.0)	12,180 (24.9)	33,093 (26.4)	50,535 (26.8)	25,308 (27.1)	22,378 (26.5)	21,653 (26.7)	22,911 (26.5)	21,527 (27.0)
None of the above	75,424 (17.8)	20,910 (22.4)	19,725 (17.0)	20,362 (15.3)	14,427 (1.7)	10,704 (17.3)	6036 (12.4)	20,307 (16.2)	38,376 (20.4)	16,544 (17.7)	14,285 (16.9)	14,178 (17.5)	15,024 (17.4)	15,393 (19.3)
Townsend deprivation, mean (SD)	−1.36 (3.06)	−0.92 (3.25)	−1.45 (3.01)	−1.58 (2.95)	−1.40 (3.02)	−1.14 (3.15)	−1.19 (3.14)	−1.42 (3.04)	−1.44 (3.01)	−0.90 (3.27)	−1.45 (3.00)	−1.58 (2.92)	−1.64 (2.90)	−1.29 (3.09)
Townsend deprivation index quartiles														
Q1 (lowest)	106,212 (25.0)	20,129 (21.5)	29,759 (25.7)	35,562 (26.8)	20,762 (25.1)	14,232 (23.0)	11,680 (23.9)	32,385 (25.8)	47,915 (25.4)	20,285 (21.7)	21,580 (25.6)	21,516 (26.5)	23,309 (27.0)	19,523 (24.5)
Q2	106,166 (25.0)	21,511 (23.0)	29,219 (25.2)	34,424 (25.9)	21,013 (25.5)	15,010 (24.2)	11,668 (23.9)	31,288 (24.9)	48,200 (25.6)	20,942 (22.4)	21,448 (25.4)	21,178 (26.1)	23,015 (26.7)	19,583 (24.6)
Q3	106,173 (25.0)	23,525 (25.2)	29,137 (25.1)	33,064 (24.9)	20,447 (24.8)	15,560 (25.1)	12,331 (25.2)	31,156 (24.8)	47,126 (25.0)	23,521 (25.2)	21,228 (25.2)	20,389 (25.1)	21,320 (24.7)	19,714 (24.8)
Q4 (highest)	106,174 (25.0)	28,338 (30.3)	27,794 (24.0)	29,701 (22.4)	20,340 (24.6)	17,190 (27.7)	13,173 (27.0)	30,578 (24.4)	45,232 (24.0)	28,559 (30.6)	20,135 (23.9)	18,035 (22.2)	18,673 (21.6)	20,772 (26.1)
Types of most commonly consumed coffee														
Coffee drinkers, n (%)	331,222 (78.0)	NA	115,909 (100.0)	132,751 (100.0)	82,562 (100.0)	52,278 (84.3)	43,979 (90.0)	104,787 (83.6)	130,178 (69.1)	43,791 (46.9)	70,465 (83.5)	63,568 (78.4)	79,968 (92.6)	73,430 (92.3)
Decaffeinated, n (%)	63,810 (19.3)	NA	22,792 (19.7)	24,755 (18.6)	16,263 (19.7)	10,353 (19.8)	7338 (16.7)	19,553 (18.7)	26,566 (20.4)	8509 (19.4)	12,579 (17.9)	11,760 (18.4)	16,234 (20.3)	14,728 (20.1)
Instant, n (%)	185,657 (56.1)	NA	59,075 (51.0)	73,168 (55.1)	53,414 (64.7)	31,757 (60.7)	24,044 (54.7)	57,380 (54.8)	72,476 (55.7)	20,701 (47.3)	37,121 (52.7)	35,252 (19.0)	46,250 (57.8)	46,333 (63.1)
Ground, n (%)	75,748 (22.9)	NA	30,729 (26.5)	33,009 (24.9)	12,010 (14.5)	9371 (17.9)	11,910 (27.1)	26,030 (24.8)	28,437 (21.8)	13,249 (30.3)	19,289 (27.4)	15,497 (20.5)	16,341 (20.4)	11,372 (15.5)
Other, n (%)	6007 (1.8)	NA	3313 (2.9)	1819 (1.4)	875 (1.1)	797 (1.5)	687 (1.6)	1824 (1.7)	2699 (2.1)	1332 (3.0)	1476 (2.1)	1059 (17.6)	1143 (1.4)	997 (1.4)

Abbreviations: A, Advanced; AS, Advanced Subsidiary; BMI, body mass index (calculated as weight in kilograms divided by height in meters squared); GCSE, General Certificate of Secondary Education; NA, not applicable; No., number; O, Ordinary; Q, quartiles; SD, standard deviation; UK Biobank, United Kingdom Biobank.

**Table 2 nutrients-14-04039-t002:** Coffee and tea consumption in relation to asthma risk: UK Biobank.

Characteristics	Hazard Ratio for Asthma							
	HR (95%CI) from Crude Model ^a^	*p*-Value	HR (95%CI) from Model 1 ^b^	*p*-Value	HR (95%CI) from Model 2 ^c^	*p*-Value	HR (95%CI) from Model 3 ^d^	*p*-Value
Coffee (cups/day)								
0	Ref		Ref		Ref		Ref	
0.5 to 1	0.872 (0.822~0.925)	<0.001	0.849 (0.800~0.901)	<0.001	0.912 (0.859~0.967)	0.002	0.919 (0.866~0.976)	0.006
2 to3	0.817 (0.771~0.866)	<0.001	0.800 (0.755~0.848)	<0.001	0.855 (0.807~0.907)	<0.001	0.877 (0.826~0.931)	<0.001
≥4	0.940 (0.882~1.001)	0.054	0.944 (0.886~1.005)	0.072	0.932 (0.874~0.993)	0.03	0.970 (0.908~1.037)	0.376
	P_trend_ = 0.002		P_trend_ = 0.003		P_trend_ = 0.002		P_trend_ = 0.090	
Tea (cups/day)								
0	Ref		Ref		Ref		Ref	
0.5 to 1	0.822 (0.755~0.894)	<0.001	0.827 (0.760~0.900)	<0.001	0.891 (0.819~0.970)	0.008	0.889 (0.816~0.968)	0.007
2 to3	0.870 (0.815~0.930)	<0.001	0.856 (0.801~0.914)	<0.001	0.935 (0.875~1.000)	0.049	0.929 (0.868~0.995)	0.037
≥4	0.870 (0.815~0.930)	0.195	0.938 (0.882~0.997)	0.041	1.000 (0.940~1.064)	0.996	0.991 (0.928~1.059)	0.799
	P_trend_ = 0.588		P_trend_ = 0.701		P_trend_ = 0.248		P_trend_ = 0.417	
Coffee—caffeinated only (cups/day)								
0	Ref		Ref		Ref		Ref	
0.5 to 1	0.837 (0.786~0.892)	<0.001	0.820 (0.770~0.870)	<0.001	0.876 (0.822~0.933)	<0.001	0.883 (0.829~0.941)	<0.001
2 to3	0.796 (0.749~0.846)	<0.001	0.911 (0.851~0.975)	<0.001	0.837 (0.786~0.890)	<0.001	0.858 (0.806~0.915)	<0.001
≥4	0.901 (0.842~0.965)	0.003	0.911 (0.851~0.975)	0.007	0.896 (0.836~0.960)	0.002	0.933 (0.868~1.003)	0.059
	P_trend_ < 0.001		P_trend_ < 0.001		P_trend_ < 0.001		P_trend_ = 0.006	
Tea and coffee—Caffeine intake quintiles								
Q1(≤160.0)	Ref		Ref		Ref			
Q2(160.0–235.0)	0.889 (0.832~0.949)	<0.001	0.866 (0.811~0.925)	<0.001	0.899 (0.842~0.961)	0.002	NA	
Q3(235.0–305.0)	0.922 (0.863~0.985)	0.016	0.898 (0.840~0.959)	0.001	0.926 (0.867~0.990)	0.023	NA	
Q4 (305.0–390.0)	0.940 (0.882~1.003)	0.061	0.915 (0.858~0.976)	0.007	0.940 (0.881~1.003)	0.062	NA	
Q5 (≥390.0)	1.035 (0.971~1.103)	0.295	1.037 (0.972~1.105)	0.274	0.999 (0.936~1.066)	0.979	NA	
	P_trend_ = 0.161		P_trend_ = 0.168		P_trend_ = 0.719			

^a^ unadjusted (crude) model; ^b^ Adjusted by age and gender; ^c^ Adjusted by age, gender, race, BMI, smoking status, education, and Townsend Index; ^d^ Adjusted by age, gender, race, body mass index, smoking status, education, and Townsend Index, and adjusted for coffee in tea analysis or for tea in coffee analysis. Abbreviations: HR: hazard ratio; CI, confidence interval; Q, quartiles.

## Data Availability

Data from the UK Biobank cannot be shared publicly, however, data are available from the UK Biobank Institutional Data Access/Ethics Committee (contact via http://www.ukbiobank.ac.uk/ (accessed on 12 September 2022) or contact by email at access@ukbiobank.ac.uk) for researchers who meet the criteria for access to confidential data.

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
