# Peer review of "Association of Coffee and Tea Consumption with the Risk of Asthma: A Prospective Cohort Study from the UK Biobank"

_nutrients, 2022, doi:10.3390/nu14194039_

Round 1
Reviewer 1 Report
This study examined associations between self-reported intake of tea and coffee and with risk of incident asthma in a total of 424,725 participants aged from 39 to 73 years old from the UK Biobank. Cox proportional hazards models were used to estimate the associations between coffee/tea consumption and incident adult-onset asthma, adjusting for age, gender, race, smoking status, body mass index (BMI), education, and Townsend deprivation index. The results showed that light to moderate coffee and tea consumption (0.5-3 cups/day) was associated with a reduced risk of adult-onset asthma, and controlling total caffeine intake from coffee and tea for a moderate caffeine dose of 160.0 to 305.0 mg/day may be protective against adult-onset asthma. There are some concerns as listed in the following:
#P1/17
* Correspondence -> # Correspondence
used to estimating -> used to estimate
*adjusting for age, sex, race, smoking status, body mass index (BMI), qualification, and Townsend deprivation index. -> adjusting for age, sex, race, smoking status, body mass index (BMI), education, and Townsend deprivation index.
* 0.5 to 1 cups/d (HR 0.858, 95% CI 0.806-0.915) -> 2 to 3 cups/d (HR 0.858, 95% CI 0.806-0.915)
#P2/17
Exposure assessment -> Exposure Assessment
#P3/17
Statistical analyses -> Statistical Analyses
#P8/17
Nonlinear association
Coffee intake, tea intake, and combined caffeine intake with asthma risk
#P12/17
Subgroup and Sensitivity analyses
#P13/17
*Figure 4: (B)tea, (D)caffeinated coffee, and (D) combination caffeine intake -> (B)tea, (C)caffeinated coffee, and (D) combination caffeine intake
#P14/17
Wee et.al[12].
#P15/17
children-and adolescents-onset
UK biobank Resource
#P15/17 Reference
Ref. 3: J Clin Invest 2022, 132(12) no page number. [J Clin Invest. 2022 Jun 15;132(12):e149371. doi: 10.1172/JCI149371.]
Ref. 12: Analysis of the Relationship between Asthma… -> Analysis of the relationship between asthma… also check Ref 10, 16, 27, 36, 37
Ref.12: Int J Environ Res Public Health 2020, 17(20) no page number. [Int J Environ Res Public Health. 2020 Oct 14;17(20):7471. doi: 10.3390/ijerph17207471.]
Ref. 16: Molecules 2020, 25(19) no page number. [Molecules. 2020 Oct 5;25(19):4553. doi: 10.3390/molecules25194553.]
Ref. 33: Nutrients 2021, 13(9) no page number. [Nutrients. 2021 Sep 2;13(9):3088. doi: 10.3390/nu13093088.]
Author Response
Response to Reviewer 1 Comments
We are grateful for the opportunity to submit a revised version of our manuscript entitled “Association of coffee and tea consumption with the risk of asthma: A prospective cohort study from the UK Biobank” to Nutrients. We greatly appreciate the insightful and constructive comments from you and believe that the revised manuscript is improved, and hope it is now suitable for publication in Nutrients. We have made major revisions in our manuscript accordingly. Changes to the revised manuscript are marked up using the “Track Changes” function. Please find our point-by-point responses below:
Point 1:
#P1/17
* Correspondence -> # Correspondence.
used to estimating -> used to estimate
*adjusting for age, sex, race, smoking status, body mass index (BMI), qualification, and Townsend deprivation index. -> adjusting for age, sex, race, smoking status, body mass index (BMI), education, and Townsend deprivation index.
* 0.5 to 1 cups/d (HR 0.858, 95% CI 0.806-0.915) -> 2 to 3 cups/d (HR 0.858, 95% CI 0.806-0.915)
#P2/17
Exposure assessment -> Exposure Assessment
#P3/17
Statistical analyses -> Statistical Analyses
#P13/17
*Figure 4: (B)tea, (D)caffeinated coffee, and (D) combination caffeine intake -> (B)tea, (C)caffeinated coffee, and (D) combination caffeine intake
Response 1: Thank you for your comments. As suggested, we have modified the corresponding text.
Point 2:
#P8/17
Nonlinear association
Coffee intake, tea intake, and combined caffeine intake with asthma risk
#P12/17
Subgroup and Sensitivity analyses
#P14/17
Wee et.al[12].
#P15/17
children-and adolescents-onset
UK biobank Resource
Response 2: Thank you for your comments. We have revised the text as follows:
#P8/17
Nonlinear association -> Nonlinear Association
Coffee intake, tea intake, and combined caffeine intake with asthma risk -> Coffee Intake, Tea Intake, and Combined Caffeine Intake with Asthma Risk
#P12/17
Subgroup and Sensitivity analyses -> Subgroup and Sensitivity Analyses
#P14/17
Wee et.al[12]. -> Wee et al.,[12].
#P15/17
Strengths and limitations -> Strengths and Limitations
children-and adolescents-onset -> children and adolescents-onset
UK biobank Resource -> UK Biobank Resource
Point 3:
#P15/17 Reference
Ref. 3: J Clin Invest 2022, 132(12) no page number. [J Clin Invest. 2022 Jun 15;132(12):e149371. doi: 10.1172/JCI149371.]
Ref. 12: Analysis of the Relationship between Asthma… -> Analysis of the relationship between asthma… also check Ref 10, 16, 27, 36, 37
Ref.12: Int J Environ Res Public Health 2020, 17(20) no page number. [Int J Environ Res Public Health. 2020 Oct 14;17(20):7471. doi: 10.3390/ijerph17207471.]
Ref. 16: Molecules 2020, 25(19) no page number. [Molecules. 2020 Oct 5;25(19):4553. doi: 10.3390/molecules25194553.]
Ref. 33: Nutrients 2021, 13(9) no page number. [Nutrients. 2021 Sep 2;13(9):3088. doi: 10.3390/nu13093088.]
Response 3: Thank you for your comments. As suggested, We have revised the literature formatting of Ref. 10,12, 16, 27, 36, 37 and added the page number of Ref. 3, 12, 16, 33.

Reviewer 2 Report
This study by Lin et. al. highlights the associations of risk of adulthood asthma to Coffee/Tea/ total caffeine intake. The work is interesting and nicely presented. However, there are minor corrections to be considered before this study is accepted for publication:
1. Can authors discuss why the education was taken as a factor to adjust restricted cubic spine model results?
2. In Fig.2, the labels of axes are hard to read.
3. In Fig.3, X and Y axes should be labelled for a better and quick readability.
4. Figure 4 is completely hard to read.
Author Response
Response to Reviewer 2 Comments
We are grateful for the opportunity to submit a revised version of our manuscript entitled “Association of coffee and tea consumption with the risk of asthma: A prospective cohort study from the UK Biobank” to Nutrients. We greatly appreciate the insightful and constructive comments from you and believe that the revised manuscript is improved, and hope it is now suitable for publication in Nutrients. We have made major revisions to our manuscript accordingly. Changes to the revised manuscript are marked up using the “Track Changes” function. Please find our point-by-point responses below:
Point 1: Can authors discuss why the education was taken as a factor to adjust restricted cubic spine model results?
Response 1: Thank you for your comments.
On Discussion (page 16), we have added the following content:
In addition to the factors previously discussed, educational background and Townsend deprivation index are socioeconomic factors that exert effects on environmental allergen exposure[41,42] and may lead to variations in coffee and tea consumption [43,44]. Sim-ilar to other studies [7,25,45], we adjusted for education and Townsend deprivation index in this study.
- Zhang Y, Yang H, Li S, Li WD, Wang Y: Consumption of coffee and tea and risk of developing stroke, dementia, and post-stroke dementia: A cohort study in the UK Biobank. PLoS Med 2021, 18(11):e1003830.
- Zhou A, Hypponen E: Long-term coffee consumption, caffeine metabolism genetics, and risk of cardiovascular disease: a prospective analysis of up to 347,077 individuals and 8368 cases. Am J Clin Nutr 2019, 109(3):509-516.
- Jabre NA, Keet CA, McCormack M, Peng R, Balcer-Whaley S, Matsui EC: Material Hardship and Indoor Allergen Exposure among Low-Income, Urban, Minority Children with Persistent Asthma. J Community Health 2020, 45(5):1017-1026.
- Zhu Y, Jing D, Liang H, Li D, Chang Q, Shen M, Pan P, Liu H, Zhang Y: Vitamin D status and asthma, lung function, and hospitalization among British adults. Front Nutr 2022, 9:954768.
- Loftfield E, Freedman ND, Dodd KW, Vogtmann E, Xiao Q, Sinha R, Graubard BI: Coffee Drinking Is Widespread in the United States, but Usual Intake Varies by Key Demographic and Lifestyle Factors. J Nutr 2016, 146(9):1762-8.
- Vieux F, Maillot M, Rehm CD, Drewnowski A: Tea Consumption Patterns in Relation to Diet Quality among Children and Adults in the United States: Analyses of NHANES 2011-2016 Data. Nutrients 2019, 11(11):2635.
- Creed JH, Smith-Warner SA, Gerke TA, Egan KM: A prospective study of coffee and tea consumption and the risk of glioma in the UK Biobank. Eur J Cancer 2020, 129:123-131.
Point 2: In Fig.2, the labels of axes are hard to read.
Response 2: Thank you for your comments. We have increased the size of the labels of axes in Figure 2 for clarity.
Point 3: In Fig.3, X and Y axes should be labelled for a better and quick readability.
Response 3: Thank you for your comments. We have labeled X axes as “HR (95 CI%) and Y axes as “Groups”
Point 4: Figure 4 is completely hard to read.
Response 4: Thank you for your comments. As you suggested, we improved the readability of Figure 4 and increased the size of the pictures in Figure 4.
